# Absorptive K-Band Bandpass Filter Using a Balanced Recursive Structure

**Seong-Mo Moon [1]**, **Han Lim Lee [2]** and **Moon-Que Lee [3],***

1   Communication RF Research Section, Electronics and Telecommunications Research Institute (ETRI), 218 Gajeong-ro, Yuseong-gu, Daejeon 34129, Korea; smmoon@etri.re.kr
2   School of Electrical and Electronics Engineering, Chung-Ang University, 84 Heukseok-ro, Dongjak-gu, Seoul 06974, Korea; hanlimlee@cau.ac.kr
3   School of Electrical and Computer Engineering, University of Seoul, 163 Siripdae-ro, Dongdaemun-gu, Seoul 02504, Korea
*   Correspondence: mqlee@uos.ac.kr

**Abstract:** This article presents a new K-band absorptive bandpass filter (ABPF) based on a microwave balanced recursive architecture. The proposed structure was configured using two passive microwave hybrid couplers, two conventional bandpass filters (BPFs), and a recursive path control module consisting of a phase shifter and an optionally variable gain amplifier. Using the proposed structure, stable return characteristics that were insensitive to the output load variation in the passband, a reduction in standing wave due to absorption in the stopband, and potentially high reliability could be achieved. Furthermore, since the same BPFs were reused, the electrical filtering order within the given physical BPF stages could be increased effectively. The proposed architecture was verified by comparing it with the performance of the conventional two-stage cascaded BPF. The measured results showed a 3 dB passband at 280 MHz with the center frequency at 19.9 GHz and improved roll-off characteristics. Furthermore, the stopband showed the reflectionless characteristic with the return loss being better than 7 dB.

**Keywords:** microwave filter; absorptive filter; non-reflective bandpass filter; high-throughput satellite (HTS)

## 1. Introduction

With the rapidly increased demand for high data rate services, high-throughput satellite (HTS) systems have recently been considered as the key technology in the satellite industry. In an HTS system, multiple narrow spot-beams with high power are used [1–3], resulting in the requirement for high-performance and reliable filters without causing performance degradation of adjacent beam elements, as is conceptually shown in Figure 1. However, conventional filters are only matched in the passband and are fully reflective in the stopband and transition band, resulting in unwanted interferences in multi-channel operations of HTS radio frequency (RF) chains configured with complicated nonlinear devices. That is, absorptive (reflectionless) filters are highly desirable to avoid possible problems caused by out-of-band reflection combined with nonlinear devices, such as intermodulation products, gain ripple, and instability [4,5]. Although there have been previous studies on reflectionless filters based on lumped elements or microstrip resonators [4–7], these techniques need to be investigated more to be adopted for K-band applications since the self-resonance characteristics of lumped components and requirements for high Q-factor increase the difficulty of realization. For K-band filters, several structures based on microstrip line resonators [8], RF-MEMS [9] and GaAs MMIC [10] have been proposed, but they are all reflective in the stopbands. Moreover, the above-mentioned filters

are all exposed to the degradation of the return characteristics when the output load is varied. Therefore, a new K-band absorptive bandpass filter (ABPF) that can simultaneously satisfy the requirements for high-performance and reliable operation must be investigated.

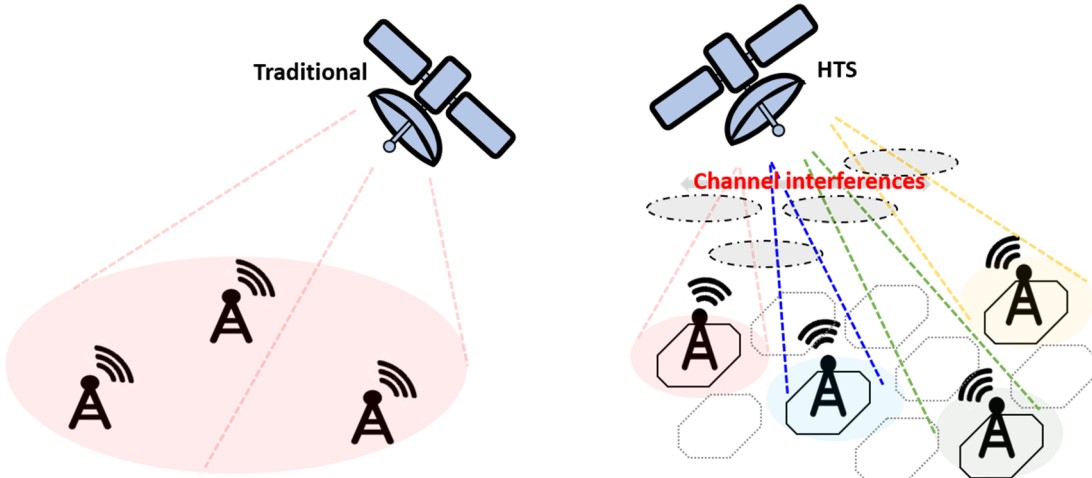

**Figure 1.** Multiple spot beams for high-throughput satellite (HTS) systems and channel interferences.

A cascaded filter structure, shown in Figure 2a, is frequently used to improve stopband rejection or to produce bandpass characteristics by combining low-pass filters (LPFs) and high-pass filters (HPFs). However, the cascaded structure using reflective filters can cause an in-band ripple or phase instability. In particular, when the output load condition is varied, the increase in reflected signal will directly appear as the input return characteristic. Furthermore, if one of the cascading elements fails, the overall performance will be degraded due to the series connection between the input and output signal chains. Thus, a new balanced ABPF structure with high reliability and performance that can overcome the conventional cascaded microwave filter was proposed in this study.

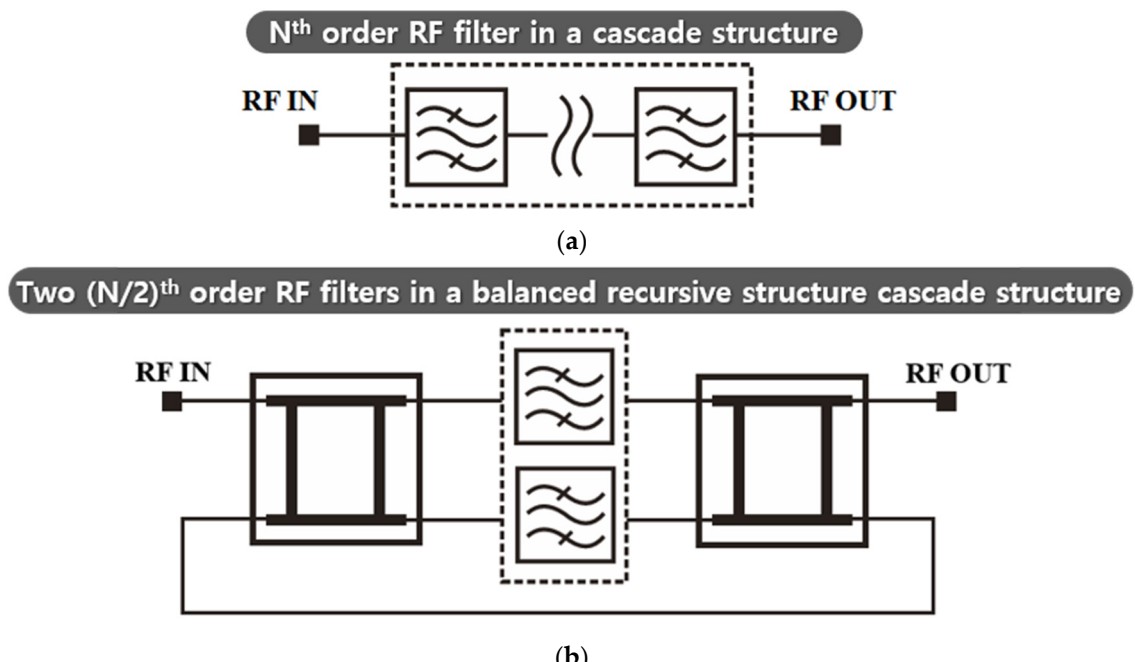

**Figure 2.** Simple schematic of multi-stage microwave filters with (**a**) a conventional cascaded structure and (**b**) the proposed balanced recursive structure. RF: radio frequency.

## 2. Design of the Proposed Structure

The balanced recursive structure, shown in Figure 2b, was proposed to enhance the reflection characteristics and reliability in the case of filter failure. The input signal is divided by the input hybrid coupler and filtered separately. Then, the filtered signal is recombined through the output hybrid coupler and fed back to the input hybrid coupler for recursive operations. Thus, a cascading effect can be achieved by the same filters without the problems of the conventional cascading method since the hybrid couplers work as buffers. Furthermore, the intrinsic nature of the balanced structure reduces the effect of output load variations since the reflected signals are returned to the isolation port of the input hybrid coupler instead of the input port. If one of the two filters is damaged, the output signal is only partially degraded in the proposed structure; both filters need to be broken simultaneously for the signal to be completely degraded. Due to the outstanding performance being based on the balanced recursive structure, the proposed structure is implemented with a phase shifter in the recursive path, as shown in Figure 3a. Here, the phase shifter adjusts the relative phase of the recursive signal to the incident signal in an orthogonal manner for each BPF to avoid interference. For the proposed structure, the reflection-type phase shifter can be the optimal choice since it has good return characteristics and does not consume additional power.

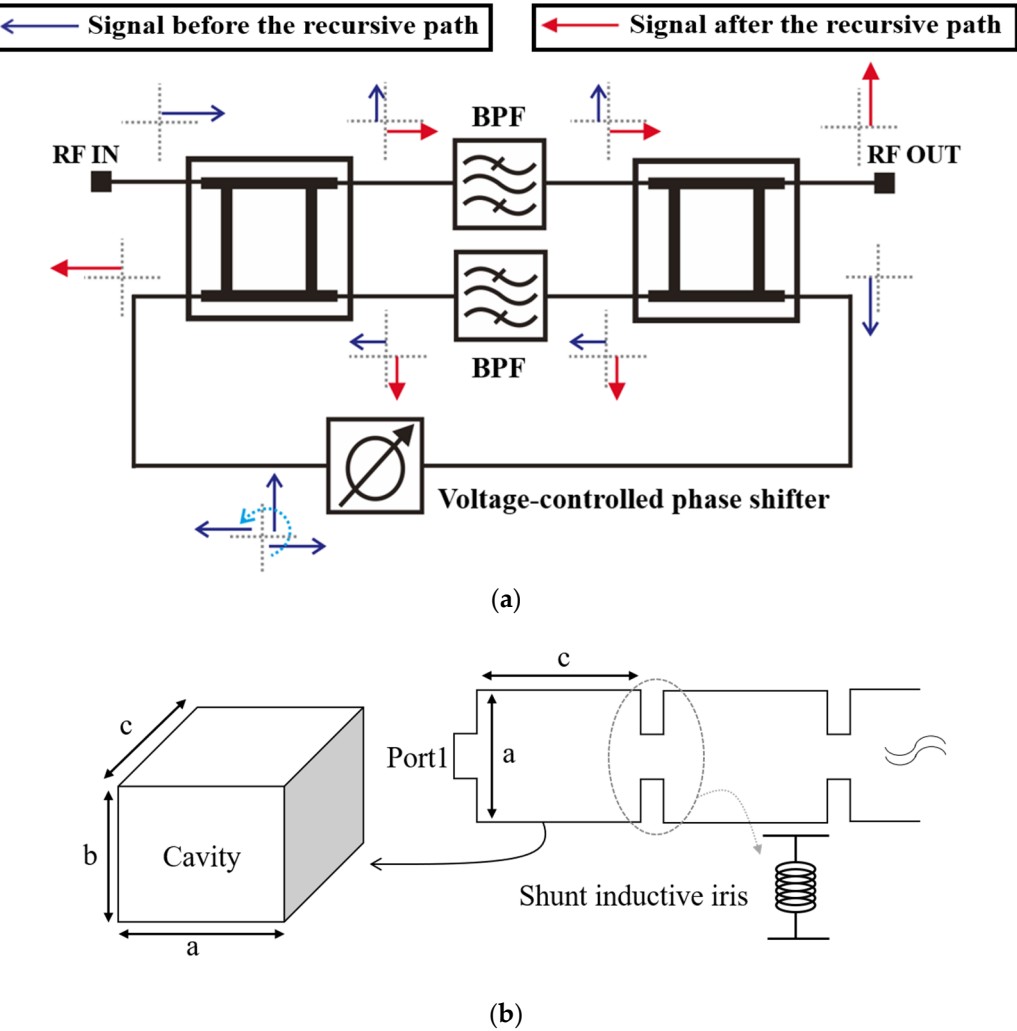

(a)

(b)

**Figure 3.** Fundamental structure of the (**a**) proposed recursive absorptive bandpass filter (ABPF) and (**b**) rectangular waveguide cavity.

Furthermore, waveguide filters were chosen for each BPF in Figure 3a due to the general advantages of their high power-handling capability and high Q-factor. The conventional rectangular waveguide geometry is shown in Figure 3b, where the cutoff wavenumber is expressed using:

$$k_c = \sqrt{\left(\frac{m\pi}{a}\right)^2 + \left(\frac{n\pi}{b}\right)^2},$$ (1)

which is used to determine the initial parameters. Here, $m$ and $n$ refer to the transverse electric $TE_{mn}$ mode of the electric wave. Then, the corresponding cutoff frequency of the rectangular waveguide can be found using:

$$f_{c,mn} = \frac{1}{2\pi \sqrt{\mu\varepsilon}} k_c,$$ (2)

where $\mu$ and $\varepsilon$ denote the permeability and permittivity, respectively. Then, the resonance wavenumber for a wave mode $TE_{mnl}$ in a waveguide cavity and the corresponding frequency can be written as Equations (3) and (4), respectively [11].

$$k_{mnl} = \sqrt{\left(\frac{m\pi}{a}\right)^2 + \left(\frac{n\pi}{b}\right)^2 + \left(\frac{l\pi}{c}\right)^2},$$ (3)

$$f_{mnl} = \frac{1}{2\pi \sqrt{\mu\varepsilon}} k_{mnl}.$$ (4)

Lastly, a fourth-order waveguide resonator cavity structure using the $TE_{01}$ mode was designed with the iris coupling, as shown in Figure 3b. The iris can be modeled as a shunt inductance [12] and the frequency characteristics of the filter follow a fourth-order Chebyshev response. Having the initial parameters calculated and modeled at a center frequency of 20 GHz within the $TE_{01}$ mode, the required dimensions were extracted and optimized by using a 3D electromagnetic (EM) simulation tool, as shown in Figure 4a. Furthermore, Figure 4b shows the 3D simulated current distribution at an arbitrary time and the operation mode.

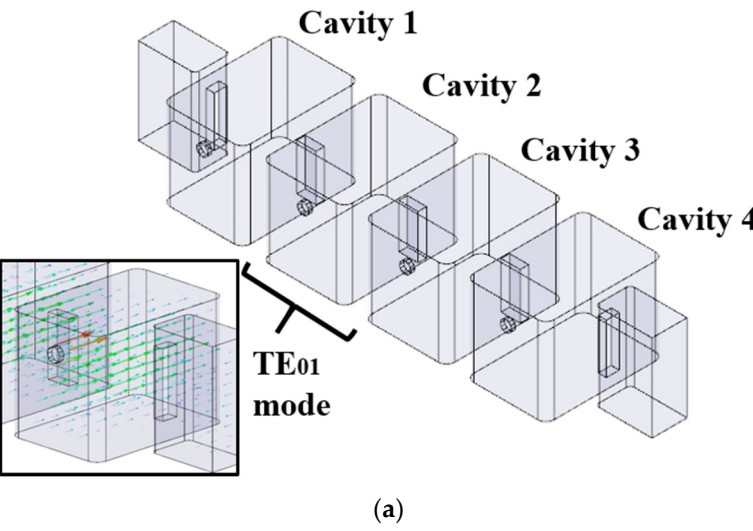

(**a**)

**Figure 4.** *Cont.*

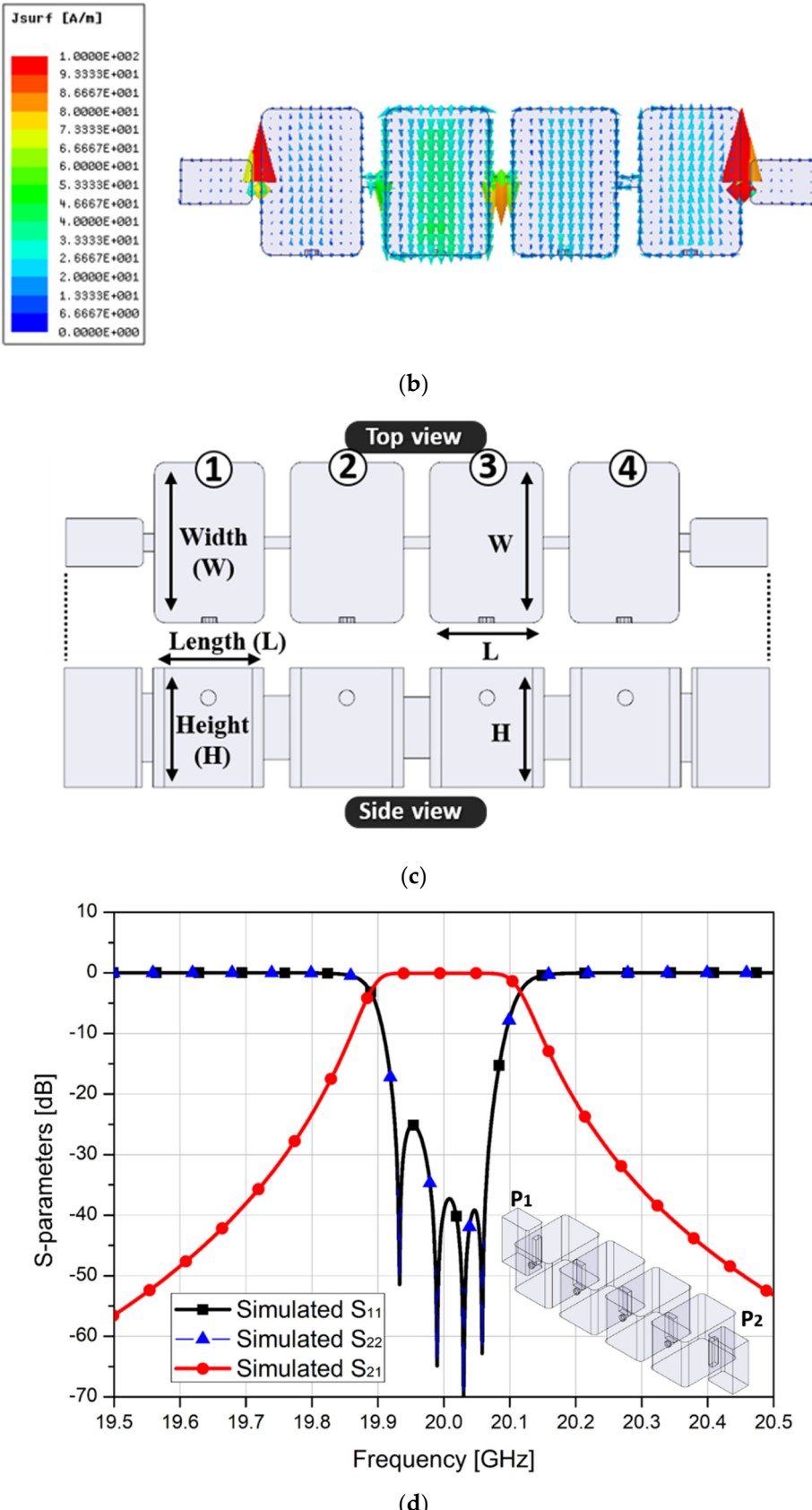

**Figure 4.** Fundamental structure of the waveguide BPF with an (**a**) overview, (**b**) geometric parameters for each cavity, (**c**) surface current distribution at an arbitrary phase, and (**d**) simulated S-parameters.

Figure 4c shows the geometric parameters for each cavity width, height, and length, where cavities 1 and 3 and cavities 2 and 4 are identical. The detailed dimensions obtained using the simulation are summarized in Table 1.

**Table 1.** Parameters of each cavity width (W), height (H), and length (L).

| Dimension | Cavity 1 | Cavity 2 | Cavity 3 | Cavity 4 |
|---|---|---|---|---|
| W (mm) | 14.50 | 14.50 | 14.50 | 14.50 |
| H (mm) | 10.67 | 10.67 | 10.67 | 10.67 |
| L (mm) | 9.80 | 10.49 | 10.49 | 9.80 |

Furthermore, the simulated S-parameters for the fourth-order cavity filter is shown in Figure 4d, where the passband insertion loss was about 0.07 dB at 20 GHz with a 3 dB bandwidth at 200 MHz.

## 3. Measured Results

Based on the simulated model, the K-band fourth-order waveguide cavity BPF and two-stage cascaded BPF were fabricated, as shown in Figure 5a,b, respectively. Referring to Figure 3a, the proposed structure requires a phase shifter in the recursive path to ensure the balance in RF signals through both BPFs. If the recursive RF signal is not phase-matched with the incoming RF signals from the input, the total RF signal to each BPF becomes unbalanced, resulting in the degradation of the filter response and absorptive characteristics. Since the high power capability and reliable performance must be ensured for a satellite payload, a reflection type phase shifter with a wideband Lange coupler was designed. The detailed characteristics of the phase shifter are well-described in [13,14] and are thus omitted from this article. Furthermore, to compensate for the loss of the input and output passive networks connected with the cavity BPFs, a variable gain amplifier was adopted in this work despite not being mandatory for configuring the proposed structure. A commercially available drive amplifier MMIC with a P1dB of 21 dBm and a gain of 17 dB at 21 GHz was used. Figure 5c shows the fabricated recursive path module, which consists of multiple phase shifters to cover 360° and a variable gain amplifier in 0.15 μm GaAs pHEMT MMICs.

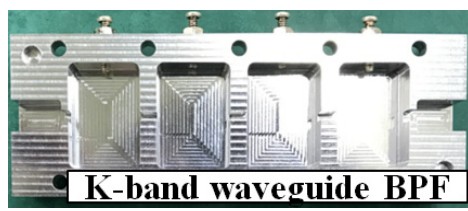

(**a**)

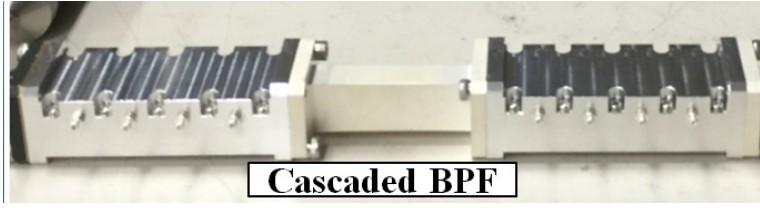

(**b**)

**Figure 5.** *Cont.*

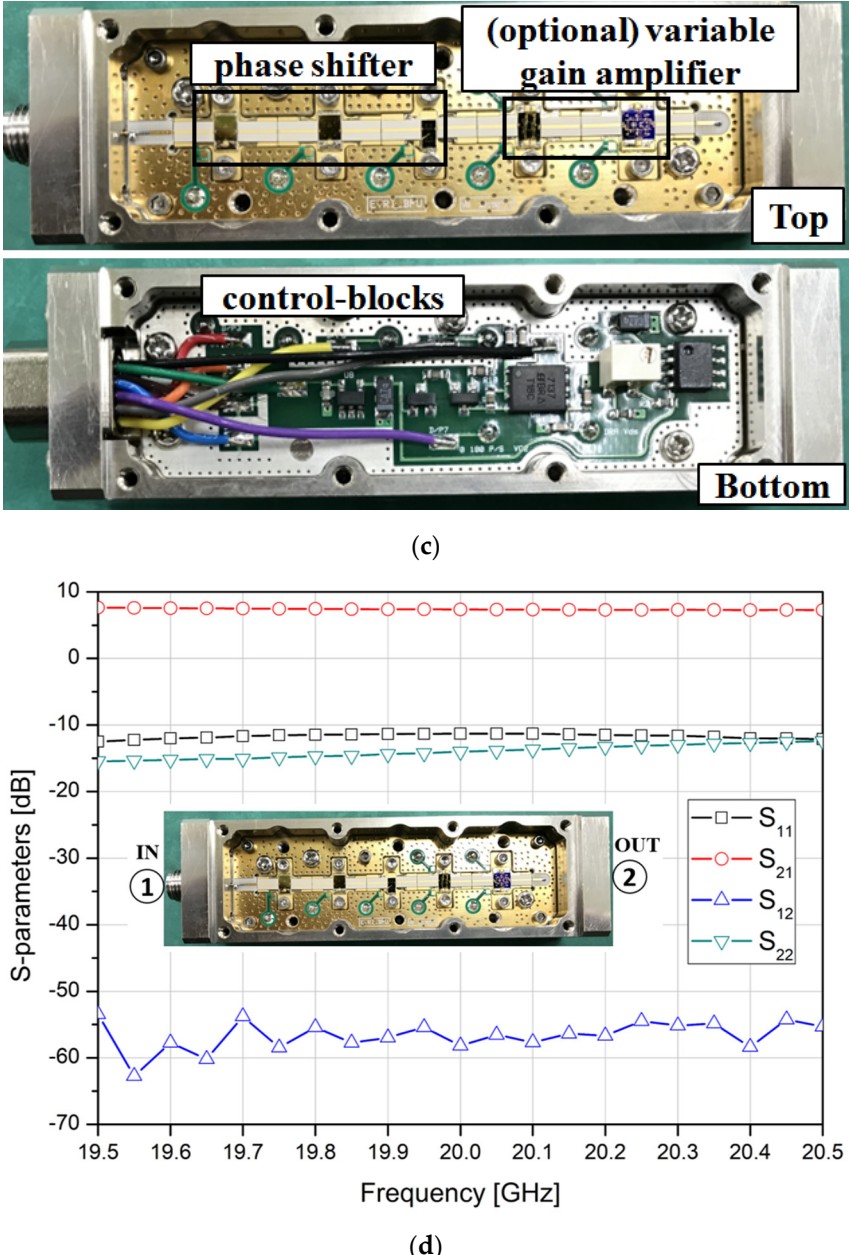

(c)

(d)

**Figure 5.** Implementation of (**a**) the single-waveguide BPF, (**b**) the cascaded BPF, and (**c**) the recursive path module and their (**d**) measured S-parameters.

Figure 5d shows the measured S-parameters of the fabricated recursive path module, where the measured input and output return losses were better than 12 dB. Furthermore, the measured transmission gain was about 7 dB, whereas the reverse isolation was higher than 55 dB, showing very stable performance within the operation band. Furthermore, the power consumption for the recursive path module depended on the amplifier showing approximately 220 mW, since the reflection type phase shifters and attenuators were based on passive operations without consuming current. Lastly, an isolator was added to the recursive path to ensure high isolation in the reverse direction during the measurement. The complete measurement setup is shown in Figure 6a. Using the single waveguide BPF as a reference, two-stage cascaded BPF and the proposed structure were measured from 19.5 GHz to 20.5 GHz, as shown in Figure 6a,b. Compared to the single BPF, both structures showed improved roll-off characteristics. However, the two-stage cascaded structure showed an unexpected peak at the stopband of 20.3 GHz, while the proposed structure showed a more stable performance. The 3 dB

passband of the proposed structure was 280 MHz, as shown in Figure 6b, and the return characteristics in the passband were better than −10 dB, as shown in Figure 6c. Furthermore, the stopband return characteristics were better than −7 dB. Therefore, for any arbitrary number of filters, the proposed recursive structure could provide improved roll-off characteristics equivalent to the cascaded structure, as well as excellent absorption characteristics in the stopband.

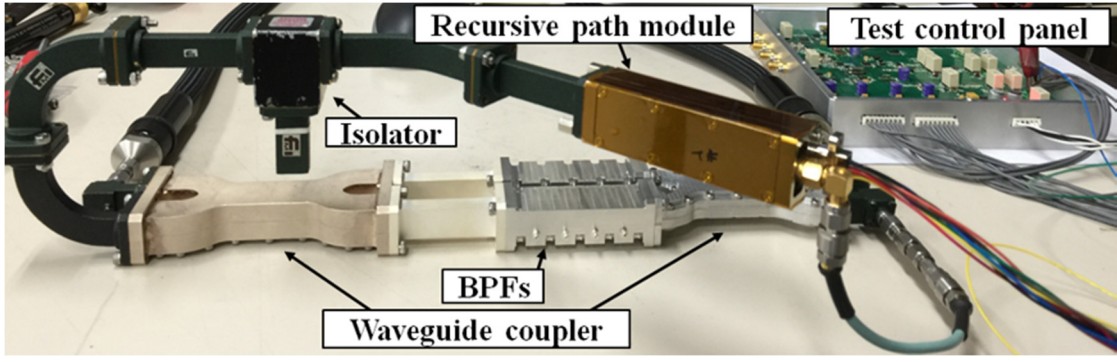

(**a**)

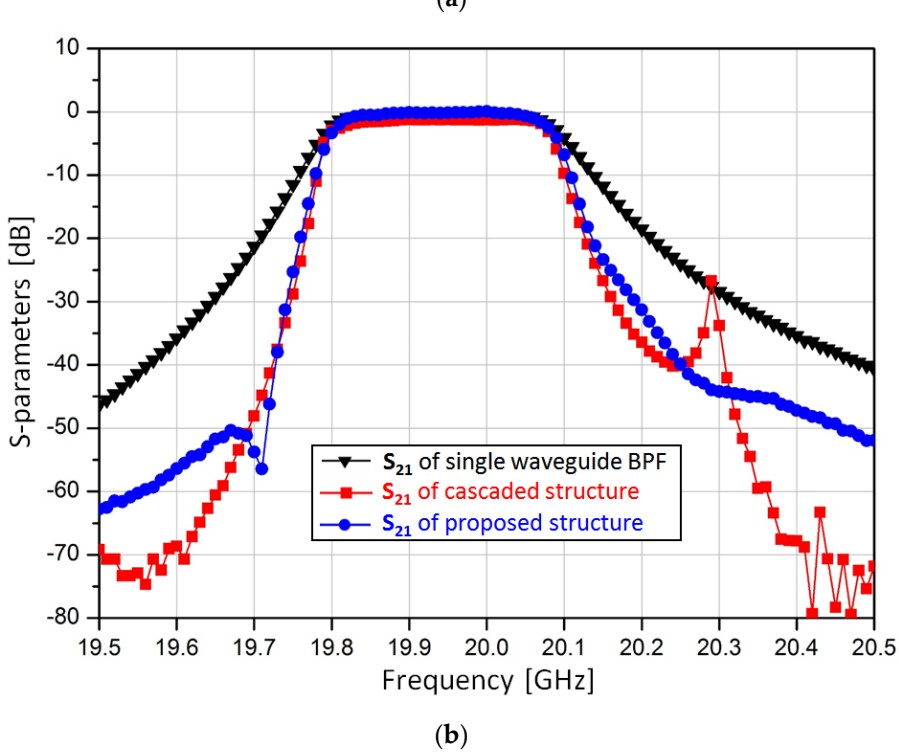

(**b**)

**Figure 6.** *Cont.*

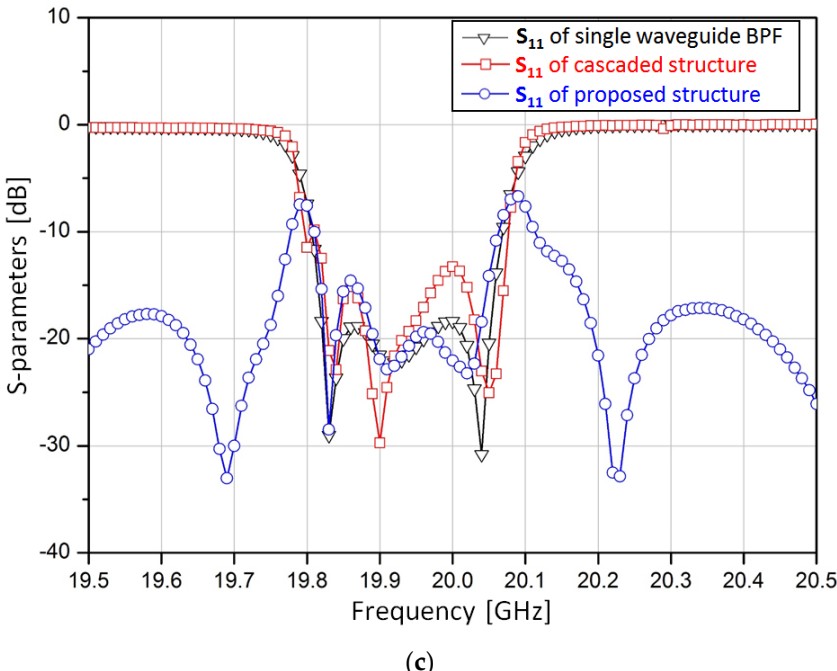

(**c**)

**Figure 6.** The proposed hybrid ABPF with (**a**) the complete measurement setup, (**b**) the measured transmission characteristics, and (**c**) the measured reflection characteristics compared with the single waveguide BPF (as a reference) and two-stage cascaded BPF.

## 4. Conclusions

In this study, a new reflectionless K-band recursive BPF that was configured using two hybrid couplers, two conventional BPFs, and a phase shifter was proposed. The proposed structure was verified using the conventional two-stage cascaded waveguide BPF, where it showed improved roll-off characteristics in the passband and good absorption characteristics in the stopband. Although the proposed structure was verified as displaying good performance with its high-quality passband, roll-off, reflection, and absorptive characteristics, the measurement was based on a prototype and only proved its electrical feasibility. The prototype has included several extra components that might be considered as bulky architecture. However, the prototype was only used to verify the structural idea, while the goal is to modify the proposed structure to provide a more refined end product using a single-chip fabrication, including all passive components and a drive amplifier. Although a cost-effective design is an important criterion, high-quality and reliable performance should be the priority for an HTS application. Thus, extended research on the possibility of single-chip fabrication for the proposed architecture is required, which is motivated by the feasibility shown in this article.

**Author Contributions:** Writing—original draft preparation, H.L.L.; investigation and validation, S.-M.M.; editing and supervision, M.-Q.L. All authors have read and agreed to the published version of the manuscript.

**Funding:** This work was supported by the 2019 Research Fund of the University of Seoul for Moon-Que Lee and the Institute for Information & Communications Technology Promotion (IITP) grant funded by the Korean Government (MSIP) (No. 2018-0-00190, Development of Core Technology for Satellite Payload) for Seong-Mo Moon.

**Conflicts of Interest:** The authors declare no conflict of interest.

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
