# Peer review of "Absorptive K-Band Bandpass Filter Using a Balanced Recursive Structure"

_electronics, doi:10.3390/electronics9101633_

Round 1

Reviewer 1 Report

The paper presents a new configuration for the design of an absorptive bandpass filter dedicated to the satellite communication in order to reduce the interferences. This idea is interesting and useful, measurement results for proof of concept are also provided.

Nevertheless the paper should be improved by providing more details about the way the different blocks have been dimensioned (or if components are from the market provide then reference should be provided as well as general configuration):

  • justify the sizes of the cavity filters,
  • has the phaser been designed by the authors? what kind of phaser is it ? I understand it is a passive phaser, but is it a component from the market? And if not, what is the passive topology of the phaser? Why did you select or design this phaser?
  • What are the properties of the variable gain amplifier? How has it been selected?

The design techniques and design difficulties must be described as they are very valuable for the readers. So the section 2 must be enlarged consequently.

Reviewer 2 Report

I have read the paper. First, there is no significance of the proposed filters is identified. There is no prior study involved. 

the proposed filter does not show the consumption of power. just a few lines that it does not consume much power is not enough to justify your claims. 

Measured results only show the description of the Figures... there is nothing related to parameters involved and other factors. 

Reviewer 3 Report

The proposed recursive k band bps is interesting even if only slightly innovative.

the following suggestion aimed at improve the quality of the work must be taken into account by the authors:

1) could you please provide a figure representing the current distribution inside the waveguide filter?

2) concerning the filter performance your design provide only a slightly improvement with respect to a standard cascade filter, but your design seems to much expensive and bulky. Could you please provide some comments in order to clarify this aspect?

Round 2

Reviewer 1 Report

Section 2 has been consequently enlarged, providing more detailed description of the system.

Author Response

There are no special comments, so We made no additional modifications.

Thank you for your review. 

Reviewer 2 Report

The author has no idea how to publish a paper in MDPI- Electronics. 

Please try to download as many of the Electronics journal paper and check the format.

1.The author itself still not clear what is the power cost of the designed filter.
2.in the design and explanation of results author just rushes towards his thoughts without any significance and application of the designed filter.
3.overall paper is just suitable for a letter, not a journal.

Author Response

Thank you for your valuable comment. The detailed response for review is listed as follows.

1. The author itself still not clear what is the power cost of the designed filter.

(Author's Reply) We already described the power consumption in the revised paper. The proposed core architecture is based on a passive operation and thus does not consume power. Just for the case where the reviewer had been curious about the optional amplifier we used in the prototype test, we already included the description as follows in the previous revision.
“Further, the power consumption for the recursive path module depends on the amplifier showing approximately 220mW, since the reflection type phase shifters and attenuators are based on passive operations without consuming current.”

2. In the design and explanation of results author just rushes towards his thoughts without any significance and application of the designed filter.

(Author's Reply) The significance has been already emphasized in the introduction by HTS applications where absorptive BPF is required. The proposed type has shown the excellent absorptive characteristic in Fig.5C with a discussion as
“Having the single waveguide BPF as a reference, two-stage cascaded BPF and the proposed structure were measured from 19.5 GHz to 20.5 GHz as shown in Figure 7 and b. Comparing to the single BPF, both structures showed the improved roll-off characteristics. However, the two-stage cascaded structure showed the unexpected peak at the stopband of 20.3 GHz while the proposed structure showed more stable performance. The 3-dB passband of the proposed structure was 280 MHz as shown in Figure 7b and the return characteristics in the passband were better than -10 dB as shown in Figure 7c. Also, the stopband return characteristics were better than -7 dB. Therefore, for any arbitrary number of filters, the proposed recursive structure can provide the improved roll-off characteristics equivalent to the cascaded structure and the excellent absorption characteristics in the stopband as well.

3. Overall paper is just suitable for a letter, not a journal.

(Author's Reply) If it is more suitable as a letter, then please change the article type to a letter, which is agreeable.